# Next-Generation Sequencing-Based Study of *Helicobacter pylori* Isolates from Myanmar and Their Susceptibility to Antibiotics

**DOI:** 10.3390/microorganisms10010196

**Published:** 2022-01-17

**Authors:** Phawinee Subsomwong, Dalla Doohan, Kartika Afrida Fauzia, Junko Akada, Takashi Matsumoto, Than Than Yee, Kyaw Htet, Langgeng Agung Waskito, Vo Phuoc Tuan, Tomohisa Uchida, Takeshi Matsuhisa, Yoshio Yamaoka

**Affiliations:** 1Department of Environmental and Preventive Medicine, Faculty of Medicine, Oita University, Yufu 879-5593, Japan; pha_203@hotmail.com (P.S.); doctordoohan@gmail.com (D.D.); kartikafauzia@oita-u.ac.jp (K.A.F.); akadajk@oita-u.ac.jp (J.A.); tmatsumoto9@oita-u.ac.jp (T.M.); langgengaw@gmail.com (L.A.W.); vophuoctuandr@gmail.com (V.P.T.); 2Department of Microbiology and Immunology, Hirosaki University Graduate School of Medicine, Hirosaki 036-8562, Japan; 3Department of Public Health and Preventive Medicine, Universitas Airlangga, Surabaya 60115, Indonesia; 4Institute of Tropical Disease, Universitas Airlangga, Surabaya 60115, Indonesia; 5Department of GI and HBP Surgery, No. (2) Defense Service General Hospital (1000 Bedded), Nay Pyi Taw 15013, Myanmar; drthanthanyee@gmail.com; 6Department of GI and HBP Surgery, No. (1) Defense Service General Hospital (1000 Bedded), Mingaladon, Yangon 11021, Myanmar; drkyawhtet@gmail.com; 7Department of Endoscopy, Cho Ray Hospital, Ho Chi Minh 749000, Vietnam; 8Department of Molecular Pathology, Faculty of Medicine, Oita University, Yufu 879-5593, Japan; tomohisa@oita-u.ac.jp; 9Department of Gastroenterology, Nippon Medical School Tama Nagayama Hospital, Tama 206-8512, Japan; matuhisa@m8.dion.ne.jp; 10Department of Medicine, Gastroenterology and Hepatology Section, Baylor College of Medicine, Houston, TX 77030, USA; 11Global Oita Medical Advanced Research Center for Health (GO-MARCH), Yufu 879-5593, Japan

**Keywords:** *Helicobacter pylori*, antibiotic resistance, infectious disease, mutations, next-generation sequencing, amoxicillin, clarithromycin, levofloxacin

## Abstract

Evaluation of *Helicobacter pylori* resistance to antibiotics is crucial for treatment strategy in Myanmar. Moreover, the genetic mechanisms involved remain unknown. We aimed to investigate the prevalence of *H. pylori* infection, antibiotic resistance, and genetic mechanisms in Myanmar. One hundred fifty patients from two cities, Mawlamyine (*n* = 99) and Yangon (*n* = 51), were recruited. The prevalence of *H. pylori* infection was 43.3% (65/150). The successfully cultured *H. pylori* isolates (*n* = 65) were tested for antibiotic susceptibility to metronidazole, levofloxacin, clarithromycin, amoxicillin, and tetracycline by Etest, and the resistance rates were 80%, 33.8%, 7.7%, 4.6%, and 0%, respectively. In the multidrug resistance pattern, the metronidazole–levofloxacin resistance was highest for double-drug resistance (16/19; 84.2%), and all triple-drug resistance (3/3) was clarithromycin–metronidazole–levofloxacin resistance. Twenty-three strains were subjected to next-generation sequencing to study their genetic mechanisms. Interestingly, none of the strains resistant to clarithromycin had well-known mutations in 23S rRNA (e.g., A2142G, A2142C, and A2143G). New type mutation genotypes such as *pbp1-A* (e.g., V45I, S/R414R), 23S rRNA (e.g., T248C), *gyrA* (e.g., D210N, K230Q), *gyrB* (e.g., A584V, N679H), *rdxA* (e.g., V175I, S91P), and *frxA* (e.g., L33M) were also detected. In conclusion, the prevalence of *H. pylori* infection and its antibiotic resistance to metronidazole was high in Myanmar. The *H. pylori* eradication regimen with classical triple therapy, including amoxicillin and clarithromycin, can be used as the first-line therapy in Myanmar. In addition, next-generation sequencing is a powerful high-throughput method for identifying mutations within antibiotic resistance genes and monitoring the spread of *H. pylori* antibiotic-resistant strains.

## 1. Introduction

The successful attempt to isolate and culture the gram-negative microaerophilic bacterium that infects the gastric epithelium has become an important event in the extensive study of *Helicobacter pylori* (*H. pylori*) [1]. The prevalence of *H. pylori* infection varies according to geographical region [2]. The interaction between *H. pylori*, the host, and the environment (e.g., socioeconomic status, dietary habits, sanitary conditions, lifestyle), and the balance among these factors is the probable cause of the geographical differences in prevalence. Currently, *H. pylori* infection is estimated to affect half of the world’s population, and it has been suggested that low prevalence has been reported only in some developed countries [2,3]. As *H. pylori* infection has been attributed to poor sanitation, poor socioeconomic status, and overcrowding, no declining tendency has been noted in the majority of developing countries [4,5].

*H. pylori* can be eradicated using antibiotics. However, the combination of more than one antibiotic with either a proton pump inhibitor or bismuth is necessary to achieve a cure rate of more than 90% [6]. *H. pylori* eradication continues to become a challenge for clinicians because of the development of antibiotic resistance. The increase in *H. pylori* resistance to clarithromycin (CAM), metronidazole (MNZ), and levofloxacin (LVX) has been reported to be responsible for the decrease in cure rates worldwide [7]. The success of individual therapies mainly depends on information on the resistance patterns in a given region. Population-based antibiotic susceptibility test data may provide valuable information for the development of an eradication strategy. In addition, understanding the underlying mechanisms of these antibiotics is also important [8]. Multiple antibiotics are used in the *H. pylori* treatment regimen, and each patient has a specific target. Investigation of the whole-genome sequence by next-generation sequencing enables researchers to comprehensively and precisely identify mutations in multiple genes simultaneously, and molecular studies to detect the mutations in *H. pylori* drug resistance-related genes play an important role in rapidly investigating possible *H. pylori* resistance statuses and provide further insights into antibiotic resistance mechanisms. 

Myanmar is a developing country located in Southeast Asia with a total population of approximately 51.5 million, with most residents living in rural areas (https://myanmar.unfpa.org/, 23 January 2021). Our previous study, conducted in Yangon and Mandalay, showed that the overall prevalence of *H. pylori* infection was 48.0% [9]. There was a report of *H. pylori* MNZ, ciprofloxacin, and LVX resistance in Myanmar [10]. However, the mechanism of antibiotic resistance, especially gene mutations, is unknown. In this study, we aimed to analyze the genetic mutations associated with the five common antibiotics used for *H. pylori* therapy in Myanmar using whole-genome sequencing techniques.

## 2. Materials and Methods

### 2.1. Study Participants

We performed an upper endoscopic survey of patients with dyspeptic symptoms in Mawlamyine (13–15 February 2017) and Yangon (16–17 February 2017). Patients with a history of gastric resection, antibiotic usage within 2 weeks before endoscopy, or previous *H. pylori* eradication treatment were excluded from this study. Continuous patients were recruited for this study. The biopsy specimen was taken from the lesser curvature of the antrum (approximately 2 cm from the pyloric ring). Specimens from the antrum were used for *H. pylori* culture, and the biopsy specimens for culture were placed in transport media and immediately stored at −80 °C until use. The samples were shipped under dry ice to Japan (Oita University Faculty of Medicine) where all analyses were performed. 

All participants provided written informed consent, and the study was approved by the ethics committee of No. (1) and No. (2) Defense Services General Hospital (1000 Bedded), Myanmar, and the Oita University Faculty of Medicine (No. P-10-12), Japan.

### 2.2. H. pylori Infection Status and Gastritis Histology Score

*H. pylori* culture was performed as described previously [11]. Briefly, the homogenized biopsy specimen was inoculated onto *H. pylori*-selective media (Nissui Pharmaceutical Co., Ltd., Tokyo, Japan) and incubated for up to 10 days at 37 °C under microaerophilic conditions (10% O_2_, 5% CO_2_, and 85% N_2_). *H. pylori* was then sub-cultured onto Brucella agar (Becton Dickinson, Sparks, MD, USA) supplemented with 7% horse blood (Nippon Bio-test, Tokyo, Japan) without antibiotics. *H. pylori* was determined based on the Gram staining results, bacterial morphology, and positive results for catalase, urease, and oxidase tests. Isolated strains were stored in Brucella broth (Becton Dickinson, Sparks, MD, USA) containing 10% glycerol and 10% horse serum and stored at −80 °C until use. 

### 2.3. Antibiotic Susceptibility Test

The antibiotic susceptibility to five major *H. pylori* antibiotics, amoxicillin (AMX), CAM, MNZ, LVX, and tetracycline (TCN), was measured using Etest (Biomerieux, Marcy-l’Étoile, France), as previously described [12]. Briefly, the *H. pylori* suspension was adjusted to be equivalent to 3.0 McFarland and then inoculated onto Mueller–Hinton II agar (Becton Dickinson, Sparks, MD, USA). An Etest strip was placed in the middle of each agar plate, and the plates were incubated at 37 °C under microaerophilic conditions (10% O_2_, 5% CO_2_, and 85% N_2_). The minimum inhibitory concentration (MIC) of *H. pylori* strains towards each antibiotic was determined after 72 h. The determination of the resistant strain was made by following the European Committee on Antimicrobial Susceptibility Testing (EUCAST; Clinical Breakpoint Table v. 5.0) criteria; MIC of >0.125 mg/L for AMX, >0.5 mg/L for CAM, >8 mg/L for MNZ, >1 mg/L for LVX, and >1 mg/L for TCN [13]. 

### 2.4. Molecular Analysis of H. pylori Resistant Strains

The genomic DNA of the *H. pylori* isolates was extracted using the DNeasy Blood and Tissue kit (Qiagen, Valencia, CA, USA) following the manufacturer’s recommendations, and the product was stored at −20 °C until use. Whole-genome sequencing was performed using the MiSeq platform (Illumina, San Diego, CA, USA). The MiSeq output was trimmed to remove low-quality bases and sequencing adapters using Trimmomatic version 0.39 [14]. Subsequently, we performed *de novo* assembly to obtain the contigs of each sequenced strain using the SPAdes algorithm [15]. The genes extracted from the contigs were analyzed using the BLAST algorithm [16]. The BLAST query was 23S rRNA (hp_r01), *rdxA* (hp0954), *frxA* (hp0642), *gyrA* (hp0701), *gyrB* (hp0501), *pbp-1A* (hp0597), and *rpl22* (hp1314) from the *H. pylori* 26695 reference genome (GenBank accession number AE000511.1). We aligned five randomly selected sensitive strains and the resistant strains to the reference genes of strain 26,695 using MAFFT software (https://mafft.cbrc.jp/alignment/software/; 15 September 2020). After confirming the absence of insertions or deletions leading to a frameshift mutation, all the sequences were aligned at the codon level except for the 23S rRNA using MAFFT software [17]. The aligned sequence was performed by variant call using SNP-sites with the first aligned sequence from strain 26,695, which, by default, was assigned as the reference [18]. Variants that appeared in the sensitive and resistant strains were considered normal variants. Variants that appeared only in the resistant strains were considered to be antibiotic resistance-related variants.

### 2.5. Data Analysis

Statistical analysis was performed using SPSS statistical software (version 23.0; IBM Corp., Armonk, NY, USA). We analyzed the discrete variables using the Fisher exact test, whereas continuous variables were determined using the Mann–Whitney U test, and statistical significance was determined when the *p*-value was <0.05. 

## 3. Results 

### 3.1. H. pylori Infection Status

A total of 150 patients (98 men and 53 women) were recruited from two cities, Mawlamyine (*n* = 99) and Yangon (*n* = 51). Among the patients, the Bamer ethnic group (*n* = 116) was the main population. Gastritis (*n* = 110) was the main endoscopic diagnosis followed by duodenal ulcer (*n* = 27), gastric ulcer (*n* = 6), gastroesophageal reflux disease (GERD) (*n* = 4), and gastric cancer (*n* = 3). The prevalence of *H. pylori* infection was 43.3% (65/150) according to culture. The mean age of the infected patients was 46.6 ± 12.2 years. There was no significant difference in the overall *H. pylori* prevalence between men and women. The infection rates in the two locations were similar (44.4% in Mawlamyine and 41.2% in Yangon). All diagnostic tests showed a higher positive rate in patients with gastric cancer and duodenal ulcers than in patients with gastritis. 

### 3.2. Antibiotic Susceptibility Test

We performed an antibiotic susceptibility test for five major antibiotics (AMX, CAM, LVX, TCN, and MNZ) on all *H. pylori* isolates (*n* = 65). Overall, the frequency of resistance to MNZ was the highest (80%), followed by LVX (33.8%), CAM (7.7%), and AMX (4.6%) (Figure 1). None of the *H. pylori* isolates were resistant to TCN. The MIC distribution in CAM, MNZ, and LVX showed a bimodal pattern, indicating persistent colonies in the population. Meanwhile, AMX and TCN showed a unimodal distribution, consistent with the sensitive traits in most strains.

The demographic data of the hosts possessing resistant strains were also analyzed (Table 1 and Appendix A). Although the resistance to LVX and MNZ was slightly higher in female subjects (45.0% and 85.0%, respectively), it was not significantly different (*p* values are 0.2 and 0.5, respectively). Meanwhile, the prevalence of CAM resistance was different among ethnicities (*p* < 0.03), and the highest resistance prevalence was observed in men (2/5; 40%) (Table 1). Among the two cities, slightly higher resistance rates of CAM, LVX, and AMX were observed in Yangon.

The multidrug resistance (MDR) patterns are shown in Table 2. Although the prevalence of resistance to each antibiotic was low, only eight isolates (12.3%) were sensitive to all five antibiotics, and 57 isolates (87.7%) were single-, double-, or triple-drug-resistant (53.8%, 29.2%, and 4.6%, respectively). The resistance to MNZ was highest (32/65, 49.2%) for single-drug, MNZ and LVX (16/19, 84.2%) for double-drug, and CAM + MNZ + LVX (3/3, 100%) for the triple-drug resistance pattern. Interestingly, in this study, all resistance to AMX was linked to other antibiotics (CAM, MNZ, or LVX). Hence, resistance to amoxicillin may indicate the presence of multidrug resistance. The antibiotic resistance rate was similar in Yangon and Mawlamyine (*p* > 0.05), although 66.7% of subjects infected with multidrug-resistant strains were obtained from Yangon.

### 3.3. Genetic Mutation Study

Mutations in genes related to antibiotic resistance have been reported to play an important role in antibiotic resistance mechanisms. We examined the association between antibiotic resistance and mutations in genes related to each antibiotic. A total of 23 strains (35% of the total culture-positive strain) were selected for genomic DNA extraction and whole-genome sequencing using MiSeq. The selected *H. pylori* strains consisted of five strains that were sensitive to all antibiotics (5/8), and 18 strains (18/57) with single-, double-, and triple-antibiotic resistance. A number of strains resistant to AMX- (*n* = 3), CAM (*n* = 5), MNZ(*n* = 11), and LVX (*n* = 13) underwent sequencing as described in Appendix A.

#### 3.3.1. Amoxicillin Resistance

Mutations in the *pbp1-A* gene, which encodes penicillin-binding protein A, are associated with AMX resistance [19,20]. Among the 23 strains, three strains showed resistance, and 20 showed sensitivity to AMX. Among the resistant strains, the mutations at positions V45I (2/3, 66.7%), S414R and V414R (2/3, 66.7%), D465K and D/del (2/3, 66.7%), V471M (1/3, 33.3%), and N564Y (1/3, 33.3%) were observed (Table 3). None of the AMX-sensitive strains carried any mutations in these positions compared to all five drug-sensitive strains. The point mutation at position N564Y was found only in strain MMM3, which had a higher MIC. The increase in MIC levels in AMX-resistant *H. pylori* strains has the potential to be mediated by this position.

#### 3.3.2. Clarithromycin Resistance

Among the 23 strains, five showed resistance to CAM and 18 showed sensitivity to CAM. All the strains were analyzed for the 23S rRNA gene, which is the main target of CAM mutations [21]. Of the five CAM-resistant *H. pylori* strains, three had a mutation at position T248C (Table 4). Surprisingly, a well-known mutation (A2143G) was not detected in the Myanmar strains (Figure 2). In addition, we investigated mutations in other target genes, such as the *rpl22* gene, which encodes a ribosomal protein that interacts with the 23S rRNA domain. Four CAM-resistant strains obtained from one to three point mutations (except strain MMM86 did not contain any mutation in the gene). None of the 13 sensitive strains carried any mutations in the 23S rRNA and *rpl22* genes.

#### 3.3.3. Levofloxacin Resistance

LVX is a fluoroquinolone group that inhibits DNA gyrase and topoisomerase IV. The common mechanism for levofloxacin resistance is point mutations in the quinolone resistance-determining region of *gyrA* and *gyrB* genes from DNA gyrase [22,23]. Eleven *H. pylori* strains were LVX-resistant, and 12 strains were sensitive to LVX. All 23 *H. pylori* strains were analyzed using the *gyrA* and *gyrB* gene sequences shown in Table 5. Among the 11 LVX-resistant strains, seven strains possessed an amino acid substitution at Asp-91, including D91N (5/11, 45.5%), D91G (1/11, 9.1%), and D91Y (1/11, 9.1%) (*p* > 0.05). The remaining strains did not possess a mutation in the quinolone-determining region of *gyrA* (MMM37, 44, 47, and 149), but possessed other possible novel mutations such as D210N, K230Q, A524V, and A661T, which were not observed in the five sensitive strains. Although GyrB is also the target of quinolone, only one strain possesses a *gyrB* mutation that has been reported in a previous study (S479G) [24]. Hence, the resistance might be explained by novel mutations in *gyrB,* such as A584V, N679H, M676V, V614I, and several other variants. We also obtained truncated *gyrA* in the two strains and truncated *gyrB* genes in the two strains. Most of the tested strains possessed multiple mutations with more than two mutations in *gyrA* (8/11, 72.7%) and *gyrB* (5/11, 45.5%).

#### 3.3.4. Metronidazole Resistance

Resistance to MNZ can change various mutations in several genes; nonetheless, the *rdxA* and *frxA* genes are highly associated with mutational inactivation. The *rdxA* gene encodes an oxygen-insensitive NADPH nitroreductase, and the *frxA* gene encodes NADPH flavin oxidoreductase. The combination of these two gene mutations may contribute to the resistance phenotype in MNZ resistance. We performed sequence analysis of *rdxA* and *frxA* from 13 MNZ-resistant and 10 sensitive strains (Appendix A). From the next-generation sequencing data, we could not obtain the *rdxA* gene from one strain (MMM33) because of the limitations of the analysis; therefore, we were able to analyze the *rdxA* gene sequences of 12 resistant strains. All 12 resistant strains possessed a mutation in *rdxA*, and three of them harbored mutations with an early stop codon. We observed that the point mutation at location V175I was the most common (8/12, 66.7%), followed by S91P (5/12, 41.7%) and R16H/C (4/12, 33.3%). The V175I point mutation could be the common phenotype of MNZ resistance due to either lower MIC (16 mg/L) or high MIC (more than 256 mg/L) obtained from the mutation in the rdxA gene. Meanwhile, the mutation in *frxA* was observed in 76.9% (10/13) of the resistant strains, and an early stop codon was only observed in one strain. Within the *frxA* gene, L33M was the most frequent point mutation (3/13, 23.1%). Moreover, the higher number of point mutations may not explain the increase in MIC by comparing the number of mutations in high and low MIC in both *rdxA* and *frxA* genes.

## 4. Discussion

This study revealed that the prevalence of *H. pylori* infection was considerably high in Myanmar, where more than half of the participants (53.3%) were infected with *H. pylori*. The prevalence of *H. pylori* infection was also high in subjects with more severe gastroduodenal diseases, such as gastric ulcers, GERD, and duodenal ulcers. The prevalence was similar to that in our previous study in the two biggest cities in Myanmar, Yangon and Mandalay, which reported that the overall *H. pylori* infection prevalence was 48.0% [9]. The involvement of environmental influences, such as sanitation, might be considered an important factor for *H. pylori* infection [25]. While sanitation access is estimated to be 77–84% in urban areas, the least sanitary coverage is only 48%, showing that there are considerably high disparities in sanitation between urban and rural areas [25]. Therefore, early diagnosis and treatment together with the improvement of sanitation in Myanmar will be necessary to control the high *H. pylori* infection rate.

The high prevalence of *H. pylori* infection highlights the importance of eradication strategies in Myanmar. To date, guidelines for *H. pylori* eradication in Myanmar are not available. The guidelines should be constructed according to resistance patterns because resistance prevalence varies widely according to geographical region, as reported by a previous Asia-Pacific and worldwide meta-analyses [26,27]. Even among cities, as in Yangon and Mawlamyine in this study, a different prevalence could be due to the different customs regarding antibiotic consumption and regulation [28]. Our previous study in Myanmar reported that MNZ-resistant *H. pylori* is quite common, while AMX-resistant strains are rare, and there are no CAM-resistant strains [10]. In this study, we also revealed a low prevalence of AMX and CAM resistance. These results were congruent with a recent meta-analysis showing that the prevalence of primary CAM resistance in the Southeast Asia Region (SEAR) was approximately 10%, while the AMX resistance was less than 5% [26]. These results suggest that a CAM-based first-line regimen may serve as a wise option for *H. pylori* eradication therapy in Myanmar [29]. Although the efficacy of standard triple therapy has decreased worldwide, especially due to the increase in CAM resistance [25], it is fortunate that CAM can be used in Myanmar. However, according to the yearly follow-up, there has been a significant increase in CAM resistance during the past few years [30]. Nevertheless, CAM should be carefully used while considering patient compliance to prevent the development of CAM resistance. A surveillance study for antibiotic resistance should also be routinely performed over the next several years.

The assessment of levofloxacin, metronidazole, and tetracycline is crucial as an alternative to consider when resistance to clarithromycin, as the most common triple therapy, arises. In Myanmar, the TCN resistance was zero. Therefore, AMX-containing triple therapy or TCN-containing quadruple therapy is also considered an alternative regimen to combat *H. pylori* infection in Myanmar [31]. As for second-line therapy, LVX-based triple therapy is recommended to be effective after the failure of a CAM-containing regimen [32]. However, LVX-resistant *H. pylori* is common (33.8%); therefore, bismuth-containing quadruple therapy consisting of bismuth, PPI, and a combination of two antibiotics, amoxicillin, tetracycline, or furazolidone, is utilized [31]. Another option for second-line treatment is rifabutin-based triple therapy, which also considers resistance of *Mycobacterium tuberculosis* in the population, as recommended in other guidelines [33].

This is the first study to investigate genetic mutations in *H. pylori* antibiotic-resistant isolates from clinical samples in Myanmar. Mutations in genes related to antibiotic resistance have been reported to play an important role in antibiotic resistance mechanisms, although this is not the only mechanism for antibiotic resistance [34]. Analysis using the whole-genome sequence has the potential to identify multiple genes and clearly evaluate nucleotide alterations among the strains [34,35,36]. Furthermore, detection of mutations in the *gyrA* and 23s rRNA genes has been used as an alternative method to identify resistance to CAM and LVX in *H. pylori.* This molecular detection offered a relatively faster result compared to the traditional phenotypic results [37,38]. However, the phenotypic results are the gold standard, despite being time-consuming, and their interpretation requires a trained individual [39]. In this study, the evaluation of both methods not only validated the molecular approach for antibiotic resistance detection but also provided insight into the mechanism of antibiotic resistance.

Amino acid substitution within *gyrA* has been associated with fluoroquinolone-related resistance. A previous study also reported that a natural transformation of mutation-carrying DNA to fluoroquinolone-sensitive isolates could account for resistance to fluoroquinolones [40]. Our study revealed several new mutations that could only be found in resistant strains. Meanwhile, almost all LVX-resistant isolates analyzed had an amino acid substitution at position 91 (Asp-91 to Asn or Tyr). Interestingly, we did not find any mutation at position 87, which was also reported to be strongly associated with fluoroquinolone resistance [41] and found in LVX-resistant strains from neighboring Southeast Asian countries such as Indonesia [12], Malaysia [42], and Cambodia [36]. Both mutations also appear in non-SEAR studies, such as in China and Turkey. Our results supported the previous finding that the mutation at position 91 increased the risk for resistance by 125.427 times compared to the wild type, while position 87 only increased the risk by 70.156 times [43]. In another study, mutations in locus 91 were only detected in resistant strains, whereas mutations at position 87 were found in sensitive strains [44]. Thus, antibiotic resistance diagnostic tests based on the mutation at locus 91 could be useful in the Myanmar population.

MNZ resistance is correlated with various RdxA mutations [45,46]. We found several isolates from Myanmar harboring the well-known R16H/C mutation in RdxA. Mutations in the R16 location are classified as class I mutations, which are expected to reduce the affinity of the apoprotein for the FMN cofactor [47]. However, we also found that most of the MNZ-resistant strains harbored the novel mutations V175I and S91P. The high prevalence of novel V175I and S91P suggests that these mutations play an important role in the MNZ resistance mechanism in *H. pylori* isolated from Myanmar. Metronidazole resistance is multifactorial, including the role of the *frxA* gene encoding NAD(P)H-flavin oxidoreductase [48]. However, concordant with other previous studies, the mutations found in this study were sporadic and difficult to determine [49,50]. Hence, further studies are needed to provide evidence of the involvement of these novel point mutations in the MNZ resistance mechanism.

Currently, the mechanism of AMX resistance in *H. pylori* is yet to be confirmed. However, several genetic determinants of *pbp-1* have been reported to be associated with AMX resistance, such as mutations at S414R and N562Y [51]. Interestingly, we found that both mutations at S414R and V45I appeared in two-thirds of the AMX-resistant strains analyzed in this study. Similar to our results, mutations at S414R and V45I were previously reported in AMX-resistant *H. pylori* strains isolated from Korean patients [52].

We also used our approach to detect mutations in 23S rRNA in CAM-resistant strains. Interestingly, strains resistant to CAM did not have well-known mutations in 23S rRNA, such as A to G substitution at nucleotide positions 2142 or 2143 [53] or A to C substitution at nucleotide position 2142 [21]. This finding also indicates that the use of a commercially available PCR-based kit focuses on locus 2142, 2143, 2146, and 2147, which might lead to false negative results [37,54]. It also underlines that PCR-based detection should be cautiously used according to the pattern of mutation in the location. Nevertheless, this study revealed a new mutation locus, such as the substitution of threonine with cysteine at locus 248, which was present in 60% of the resistance strains and absent in the sensitive strains. Although natural transformation is necessary to confirm, this finding and other mutations in the 23S rRNA gene might offer new insights into the antibiotic resistance mechanism. It is also indicated that antibiotic resistance is multifactorial and correlated to induce multidrug resistance, which was also prevalent in Myanmar. Mechanisms such as the increase in efflux pump expression, dormancy, and biofilm formation could be responsible [34,55,56].

This study had several limitations. First, the samples were only collected from two locations in Myanmar, so this result may not represent Myanmar as a whole country. Second, the number of isolates used for antibiotic susceptibility testing was relatively small. However, our results may serve as an important basis for deciding a general policy for *H. pylori* eradication strategies in Myanmar. Third, some genetic determinants of antibiotic resistance could not be confirmed because of the small number of isolates (e.g., only five CAM-resistant strains, three AMX-resistant strains, and no TCN-resistant strain). Fourth, despite the advantage of the whole-genome sequencing method, the sequencing error possibly affected the assembly result and yielded genes with some missing parts that could also be interpreted as a truncated gene. This phenomenon that appears in *gyrA* and *gyrB* should be cautiously addressed and requires further confirmation. Finally, further studies, such as clinical trials, are important to examine the efficacy of eradication regimen therapy.

## 5. Conclusions

The prevalence of *H. pylori* infection in Myanmar is high. Antibiotic susceptibility testing showed that the resistance towards MNZ and LVX was high, while the resistance towards AMX, CAM, and TCN was low. Antibiotic resistance mechanisms might be explained by genetic mutations within various *H. pylori* genes. The low prevalence of AMX and CAM resistance suggests that AMX- and CAM-containing therapy regimens can still be used as a first-line therapy to combat the high prevalence of *H. pylori* infection in Myanmar.

## Figures and Tables

**Figure 1 microorganisms-10-00196-f001:**
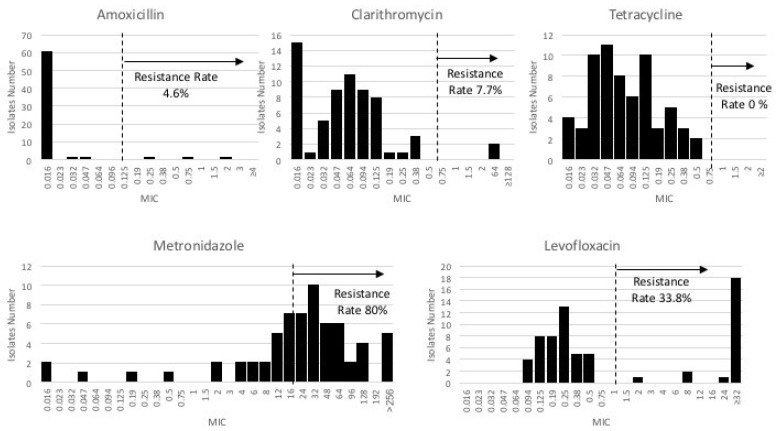
The five major antibiotic resistance rates of 65 *H. pylori* isolates from Myanmar. The clinical breakpoints were determined following the European Committee on Antimicrobial Susceptibility Testing (EUCAST; Clinical Breakpoint Table v. 5.0) criteria; MIC of >0.125 mg/L for AMX, >0.5 mg/L for CAM, >8 mg/L for MNZ, >1 mg/L for LVX, and >1 mg/L for TCN.

**Figure 2 microorganisms-10-00196-f002:**
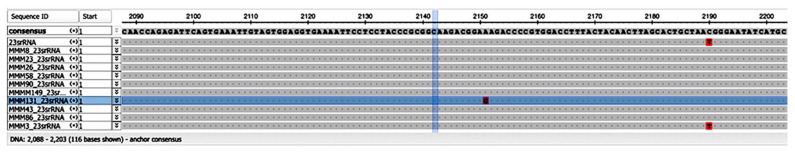
The alignment of 23S rRNA genes by MUSCLE. These region site had been reported to have mutations in the previous study. However, in Myanmar strains, mutation in the locus 2142 or 2147 was not observed.

**Table 1 microorganisms-10-00196-t001:** Distribution of antibiotic resistance based on basic characteristics.

Characteristic	*n*	Single Antibiotic Resistance (%)				
		AMX	CAM	LVX	MNZ	TCN
**Total**	65	3 (4.6)	5 (7.7)	22 (33.8)	52 (80.0)	0 (0.0)
Sex						
Male	45	3 (6.7)	4 (8.9)	13 (28.9)	35 (77.8)	0 (0.0)
Female	20	0 (0.0)	1 (5.0)	9 (45.0)	17 (85.0)	0 (0.0)
Location						
Mawlamyine	44	3 (6.8)	3 (6.8)	12(27.3)	34 (77.3)	0 (0.0)
Yangon	21	0 (0.0)	2 (9.5)	10 (47.6)	18 (85.7)	0 (0.0)
Ethnicity						
Bamer	53	2 ()	2 (3.8)	17 (32.1)	43 (81.1)	0 (0.0)
Kayen	4	0 (0.0)	1 (25.0)	2 (50.0)	4 (100)	0 (0.0)
Mon	5	1 (.0)	2 (40.0)	3 (60.0)	3 (60.0)	0 (0.0)
Others	3	0 (0.0)	0 (0.0)	0 (0.0)	2 (66.7)	0 (0.0)

**Table 2 microorganisms-10-00196-t002:** Prevalence of resistance toward single and multiple drugs in Myanmar.

Susceptibility Test Result	*n* (%)
**Number of strains**	**65**
**Sensitive**	**8 (12.3)**
**Single-drug resistance**	**35 (53.8)**
AMX	0 (0.0)
CAM	1 (1.5)
LVX	2 (3.1)
MNZ	32 (49.2)
TCN	0 (0.0)
**Double-drug resistance**	**19 (29.2)**
AMX + CAM	1 (5.3)
AMX + MNZ	1 (5.3)
AMX + LVX	1 (5.3)
MNZ + LVX	16 (84.2)
**Triple-drug resistance**	**3 (4.6)**
CAM + MNZ + LVX	3 (100)

**Table 3 microorganisms-10-00196-t003:** Mutations in *pbp-1* gene that were associated with AMX-resistance.

Gene	Strain	Sensitive (*n* = 5)	MMM3	MMM37	MMM25
*pbp1*	MIC (mg/L)		2	0.25	0.75
	Position				
	45		V/I	V/I	
	414		S/R		V/R
	465			D-Del	D/K
	471				V/M
	564		N/Y		

D: Aspartic acid, I: Isoleucine, K: Lysine, M: Methionine, N: Asparagine, R: Arginine, S: Serine, V: Valine, Y: Tyrosine, Del: Deletion, MMM: bacteria strain.

**Table 4 microorganisms-10-00196-t004:** Mutations in 23S rRNA and *rpl22* that were associated with CAM-resistance.

Gene	Strain	Sensitive (*n* = 5)	MMM3	MMM43	MMM86	MMM131	MMM149
	MIC (mg/L)		0.38	16	8	128	64
23S rRNA	Position						
	64		G/A				
	186			T/C			
	228		G/A				
	242			T-Del			
	243			T-Del			
	247		A-Ins				
	248		T/C		T/C		T/C
	425					A/G	
	513						C/T
	559						C/T
	763				A/G		
	822						G/A
	1286						A/G
	1312						C/T
	2219			C/T			
	2937					G/A	
*rpl22*	16			T/C			
	20			G/A			
	69		T/C	T/C			
	125						C/T
	135		G/A				
	147						G/T
	237			GCG-Ins		C/T	
	267						A/G

A: Alanine, C: Cysteine, G: Glycine, T: Threonine, Del: Deletion, Ins: Insertion, MMM: bacteria strain.

**Table 5 microorganisms-10-00196-t005:** Mutations in *gyrA* and *gyrB* that were associated with LVX-resistance.

Gene	Strain	MMM33	MMM37	MMM43	MMM44	MMM47	MMM54	MMM62	MMM131	MMM135	MMM145	MMM149
	MIC (mg/L)	32	>32	>32	>32	>32	>32	>32	>32	>32	2	>32
*gyrA*	Position											
	3								D/G	D/G		
	20											
	91	D/N		D/N			D/N	D/G	D/Y	D/N	D/N	
	172			V/I								
	208			G/E			G/R			G/E		
	210		D/N									
	230		K/Q									
	234										I/V	
	246										V/M	
	285											
	407	A/V						A/V	A/V			
	497			D/E								
	517						V/M					
	524		A/V								A/V	
	557			I/T								
	612											R/C
	635											R/K
	661				A/T							
	668			T/A								
	684			I/M			I/M			I/M		
	688					T/A	T/A					
	703						V/I					
	709						S/N			S/N		S/N
	712	G/S								G/S		G/S
	735											
	760									D/M		
	Truncated					1-285						1-735
*gyrB*	64							E/G				
	160										S/N	
	215									T/A		
	230							A/T				
	240								S/A			
	479								S/G			
	573				N/G							
	584	A/V										
	614									V/I	V/I	
	620							L/S				
	676		M/V									
	679	N/H										
	Truncated			1-1476								1-1476

A: Alanine, C: Cysteine, D: Aspartic acid, E: Glutamic acid, G: Glycine, H: Histidine, I: Isoleucine, K: Lysine, L: Leucine, M: Methionine, N: Asparagine, Q: Glutamine, R: Arginine, S: Serine, T: Threonine, V: Valine, Y: Tyrosine, MMM: bacteria strain.

## Data Availability

The nucleotide sequence data reported in this study are available under the DDBJ accession number LC595322-LC595338 (*gyrA*), LC595339-LC595355 (*gyrB*), LC595450-LC595468 (hp0642, *frxA*), LC595507-LC595524 (hp0954, *rdxA*), LC595573-LC595581 (*pbp*1-A), LC595582-LC595591 (*rpl22*) and LC595592-LC595601 (23s rRNA).

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
