# Peer review of "Next-Generation Sequencing-Based Study of Helicobacter pylori Isolates from Myanmar and Their Susceptibility to Antibiotics"

_microorganisms, 2022, doi:10.3390/microorganisms10010196_

Round 1
Reviewer 1 Report
This manuscript by Phawinee Subsomwong et al. describe interesting data on the WGS results of Myanmarian strains of H. pylori.
Minor modifications are needed.
Italicize bacterial names, genes, de novo and e.g.
Prefer passive voice.
Reference website as for other article
EUCAST version has to be indicated.
Why have the authors use MiSeq, that could be considered as less efficient compared to NextSeq or even non-Illumina sequencing platform.
Author Response
This manuscript by Phawinee Subsomwong et al. describe interesting data on the WGS results of Myanmarian strains of H. pylori. Minor modifications are needed. Italicize bacterial names, genes, de novo and e.g. Prefer passive voice.
Answer: According to the comment, we corrected all bacterial names, genes, de novo in italic form. The throughut sentences were checked by professional English Editing Service.
Reference website as for other article
Answer: Thank you for your comment, we added the Reference website “[ ]. European Committee on Antimicrobial Susceptibility Testing. Breakpoint tables for interpretation on MICs and zone diameters. EUCAST Clinical Breakpoint Table v. 5.0; http://www.eucast.org/clinical breakpoints” on page 13.
EUCAST version has to be indicated.
Answer: Based on the comments, we updated the EUCAST detail “EUCAST; Clinical Breakpoint Table v. 5.0” on pages 3 and 4.
Why have the authors use MiSeq, that could be considered as less efficient compared to NextSeq or even non-Illumina sequencing platform.
Answer: Thank you very much for the comment. We agree that Miseq could be considered less efficient compared to NextSeq or non-Illumina sequencing platform. However, the MiSeq could read a target higher than NextSeq (2 x 300 pb in MiSeq while 2 x 150 pb in NextSeq). In addition, the MiSeq is available in our university, so not only is cost-effective, (Purchased only the reagents) but also convenient to perform.
Reviewer 2 Report
In the present original article Subsomwong et al investigated genetic mutations conferring antibiotic resistance to H. pylori in 65 cultured strains by NGS. They found very high resistance to metronidazole and very low to clarithromycin, amoxicillin and tetracycline. Main comments:
1) The most unusual finding was that clarithromycin resistant strains did not show classical mutations (A2143G, A2143C and A2142G). regarding this last finding, where did patients live, in an urban or rural context? Patients isolation could explain such result.
2) Why only 23 out of 65 strains underwent NGS?
3) Table 3 is unclear, please explain the meaning of S, R, V, I, MM3, MMM37 and MM25
4) Please check references (e.g. n.11 has no title).
5) Table 1 “ clinical outcome” is not proper, “picture” is more fitting.
Author Response
In the present original article Subsomwong et al investigated genetic mutations conferring antibiotic resistance to H. pylori in 65 cultured strains by NGS. They found very high resistance to metronidazole and very low to clarithromycin, amoxicillin and tetracycline. Main comments:
1) The most unusual finding was that clarithromycin resistant strains did not show classical mutations (A2143G, A2143C and A2142G). regarding this last finding, where did patients live, in an urban or rural context? Patients isolation could explain such result.
Answer: We appreciate the valuable suggestions from the reviewer. We agree that where patients lived (urban/ rural) might explain the result. However, this study was investigated in two places in Myanmar (Mawlamyine and Yangon) that are located in urban area hence, our data did not sufficient to conclude that this phenomenon is due to patient`s location. Moreover, this is the first study to report the gene mutation of H. pylori strains in Myanmar so we do not have the other reference strains for comparison. From this point, we decided to check other H. pylori strains isolated from Myanmar (previous study from our group).
2) Why only 23 out of 65 strains underwent NGS?
Answer: We planned to do all 65 strains, however, our budget was enough for about ~20 samples. According to this limitation, we selected those H. pylori samples including the sensitive strains, and the strains with single-, double-, and triple-antibiotic resistance for performing the NGS. In the future, we would like to sequence all the samples.
3) Table 3 is unclear, please explain the meaning of S, R, V, I, MM3, MMM37 and MM25
Answer: According to the reviewer comment, we added the explaination of amino acid code; S, R, V, I, and bacteria strain; MMM3, MMM37, and MMM25 in Table 3. on page 7. and others in Table 4, 5, and Supplementary Table 2 on pages 8, 9, and Supplementary data.
4) Please check references (e.g. n.11 has no title).
Answer: Thank you very much, we have updated all the reference.
5) Table 1 “ clinical outcome” is not proper, “picture” is more fitting.
Answer: Thank you very much. We changed the clinical outcome to the picture shown in the “Supplementary Figure 1” in the Supplementary data and we deleted the clinical outcome from Table 1.